# The Inflection Point of Single Event Transient in SiGe HBT at a Cryogenic Temperature

**Xiaoyu Pan** [1,2], **Hongxia Guo** [2,*], **Chao Lu** [1], **Hong Zhang** [3] **and Yinong Liu** [1]

1    The Key Laboratory of Particle and Radiation Imaging, Ministry of Education, Department of Engineering Physics, Tsinghua University, Beijing 100084, China
2    State Key Laboratory of Intense Pulsed Radiation Simulation and Effect, Northwest Institute of Nuclear Technology, Xi'an 710024, China
3    The School of Material Science and Engineering, Xiangtan University, Xiangtan 411105, China
*    Correspondence: guohxnint@126.com

**Abstract:** Basing our findings on our previous pulsed laser testing results, we have experimentally demonstrated that there is an inflection point of a single event transient (SET) in the silicon-germanium heterojunction bipolar transistors (SiGe HBTs) with a decreasing temperature from +20 °C to −180 °C. Additionally, the changes in the parasitic resistivity of the carrier collection pathway due to incomplete ionization could play a key role. In this paper, we found that the incident-heavy ion's parameters could also have an important impact on the SET inflection point by introducing the ion track structures generated by Geant4 simulation to the TCAD transient simulation. Heavy ion with a low linear energy transfer (LET) will not trigger the ion shunt effect of SiGe HBT and the inflection point will not occur until −200 °C. For high LET ions' incidence, the high-density electron-hole pairs (EHPs) could significantly affect the parasitic resistivity on the pathway and lead to an earlier inflection point. The present results and methods could provide a new reference for the effective evaluation of single-event effects in bipolar transistors and circuits at cryogenic temperatures and provide new evidence of the SiGe technology's potential for applications in extreme cryogenic environments.

**Keywords:** SiGe HBT; Geant4; TCAD simulation; single event transient; cryogenic temperature

## 1. Introduction

Today, NASA is preparing to go back to the moon with Artemis missions and will build an Artemis Base Camp on the lunar surface (−180 °C ~ +120 °C). SpaceX is also making continuous efforts to land human beings on Mars (−133 °C ~ +27 °C) in Starships. All of these great missions require the support of large thrust rockets and how to improve their payload is a concern. As we all know, there are usually bulky "warm boxes" to protect the electronic systems in an extreme environment which could cause additional consumption [1]. Fortunately, SiGe HBT could be a candidate to change this situation [2].

SiGe HBT has excellent RF performance and good compatibility with silicon-based technologies, and has been widely used in wireless communication, phased-array radars, etc. [3]. Furthermore, SiGe HBT has inherent resistance to a total ionizing dose (TID) effect [4]. Meanwhile, thanks to the introduction of Ge content to the intrinsic base region, it could work over a wide temperature range (especially cryogenic temperatures) [5]. Hence, electronic systems using SiGe technologies have the potential to operate well without the "warm boxes".

However, things do not always go smoothly. SiGe HBT is sensitive to a SET and this sensitivity increases as the device feature size decreases [5]. There have been many related studies at room temperature [6–8], which can help us to understand the underlying mechanism of SiGe HBT's SET. According to the existing research, there are few studies on the impact of temperature on the SiGe HBT's SET. And generally, researchers attribute

the main cause of the SET's variation with temperature to the carrier mobility's variation such as the study on proton-induced SEU in SiGe digital logic at cryogenic temperatures in which the SET peaks increase as the temperature decreases [9]. The inflection point of the SET peaks was found by the TCAD simulation for the first time, which shows the impact of impurities' incomplete ionization (abbreviated as i.i.) at cryogenic temperature [10,11]. However, the heavy ion's LET is only 0.01 pC/μm in the simulation results that could not trigger the ion shunt effects [12] of the SiGe HBT's emitter/base/collector/substrate (E/B/C/S) stack.

As is shown in Figure 1, our previous study experimentally demonstrated the existence of the SET's inflection point for the first time by carrying out pulsed laser testing over a wide temperature range (−180 °C ~ +20 °C) [13]. We found that the change in parasitic resistance in the carrier collection pathway is an important reason for the peak inflection point. Additionally, the parasitic resistance depends on both the concentration and mobility of electrons and holes. The variation in the carrier mobility with the temperature has been studied extensively for a long time. Furthermore, we also discussed the ionization rate of intrinsic doping in our previous study. One more thing to mention so far is that we have not yet discussed the impact of the heavy-ion induced EHPs on the parasitic resistance.

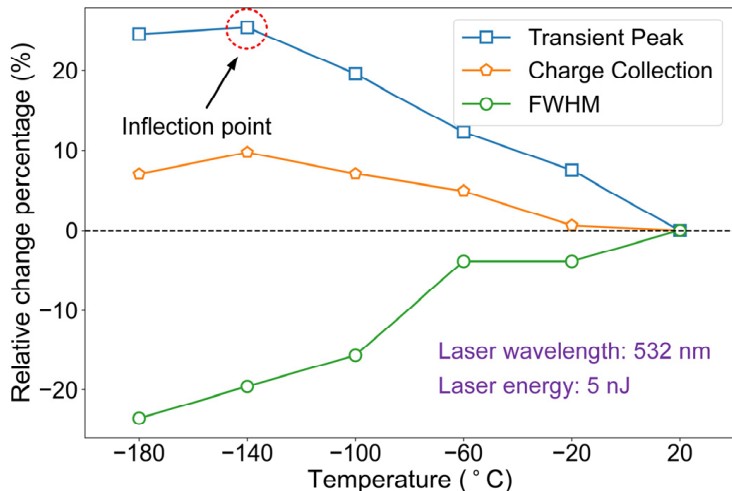

**Figure 1.** The relative percentage change (relative to 20 °C) on collector's transient peak, charge collection, and FWHM from pulsed laser testing.

In this paper, we focused on a study of the impact of the incident heavy ion's parameters on the inflection point of SET peaks. We built a simulation method which helped us to introduce the heavy-ion induced EHP's distribution generated by Geant4 calculation to the TCAD device simulation directly. When the LET value of the incident heavy ion is too low to trigger the ion shunt effect, the collector's transient current is mainly derived from collector/base (C/B) junction and collector/substrate (C/S) junction. In this case, heavy-ion induced EHPs are relatively low to the intrinsic doping and the temperature corresponding to the inflection point comes later (even up to −200 °C). As a comparison, when the LET value is relatively high and the ion shunt effect turns on at this time, then the heavy-ion induced EHPs can also have a significant impact on the total parasitic resistance of the charge collection pathway. In this case, the inflection point will come much earlier (about −160 °C).

## 2. TCAD 2-D Process Simulation

### 2.1. DUT Description

In this paper, the device under test (DUT) is a low-noise SiGe HBT (NPN transistor) provided by the School of Integrated Circuits, Tsinghua University. The DUT's lithographic node is 400 nm and the chip layout is configured as a 4E5B2C interdigital structure, shown

in Figure 2. The peak Ge content in the base region is close to 14% and has a trapezoidal distribution. The detailed device information can be found in our previous study and will not be repeated here [14].

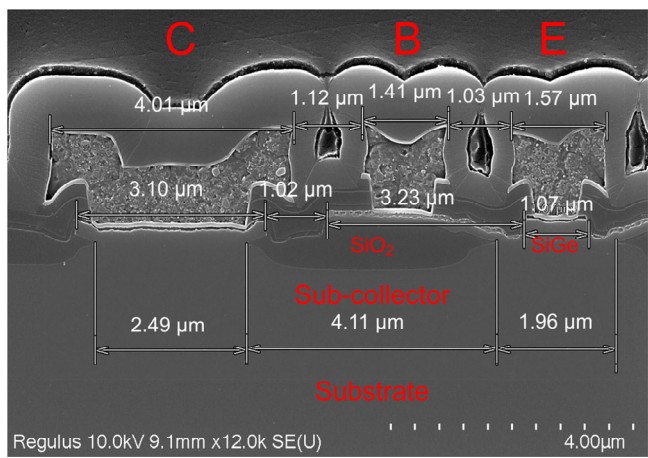

**Figure 2.** The SEM figure shows a cross-section of the DUT.

### 2.2. TCAD 2-D Process Model

In this section, we built a 2-D TCAD process model according to the DUT's production process, shown in Figure 3. In particular, this model is a simplified 1E2B2C structure to save the simulation time and achieve better simulation convergence at a cryogenic temperature. When the ion shunt effect is triggered, it can be simply understood that the EHPs generated by the incident heavy ion could build a bridge between the emitter and the collector. At this time, the total parasitic resistance of the collector's charge collection pathway includes the emitter resistance $R_E$, vertical base resistance $R_{B-V}$, selectively implanted collector (SIC) resistance $R_C$ and lateral sub-collector resistance $R_{SC}$.

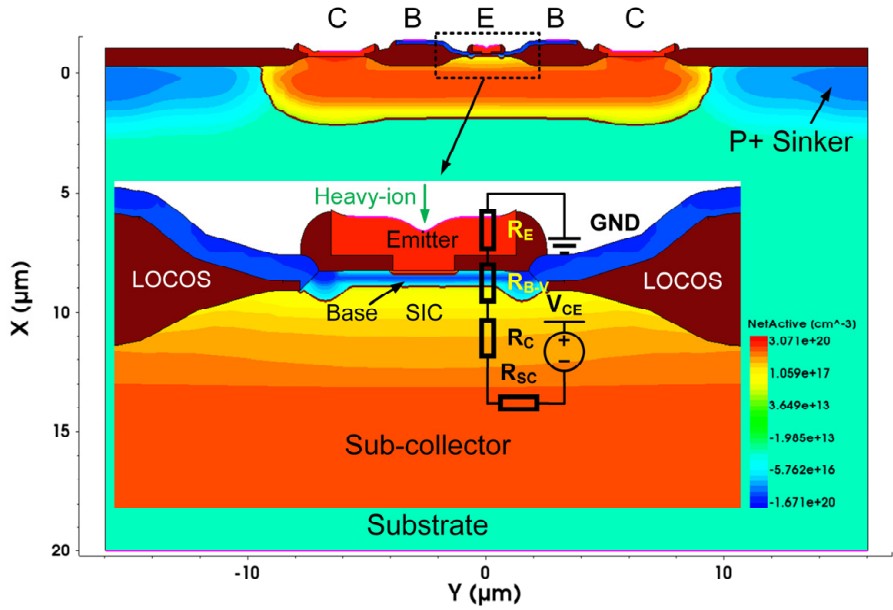

**Figure 3.** The 2-D TCAD process model with 1E2B2C structure and the inside zoom view shows the parasitic resistance of the E/B/C stack when the ion shunt effect is triggered.

### 2.3. Simulation Results

Because it is the most sensitive volume, we fixed the heavy ion's incident position at the emitter center during the whole simulation. The characteristic distance and incident

depth of the heavy ion are set to 0.2 µm and 20 µm, respectively. In addition, the device bias is defined as the cut-off bias $V_{CE}$ = 2 V, $V_{BE}$ = 0 V (or the C/S junction reverse bias). In particular, the simulation physics models are consistent with our prior study, including the incomplete ionization model [13].

We chose four temperature points from −140 °C to −200 °C with intervals of 20 °C. The simulation results are shown in Figure 4, one can see that the transient current is quite different when the LET values of the incident heavy ion are 0.01 pC/µm, 0.05 pC/µm and 0.1 pC/µm respectively.

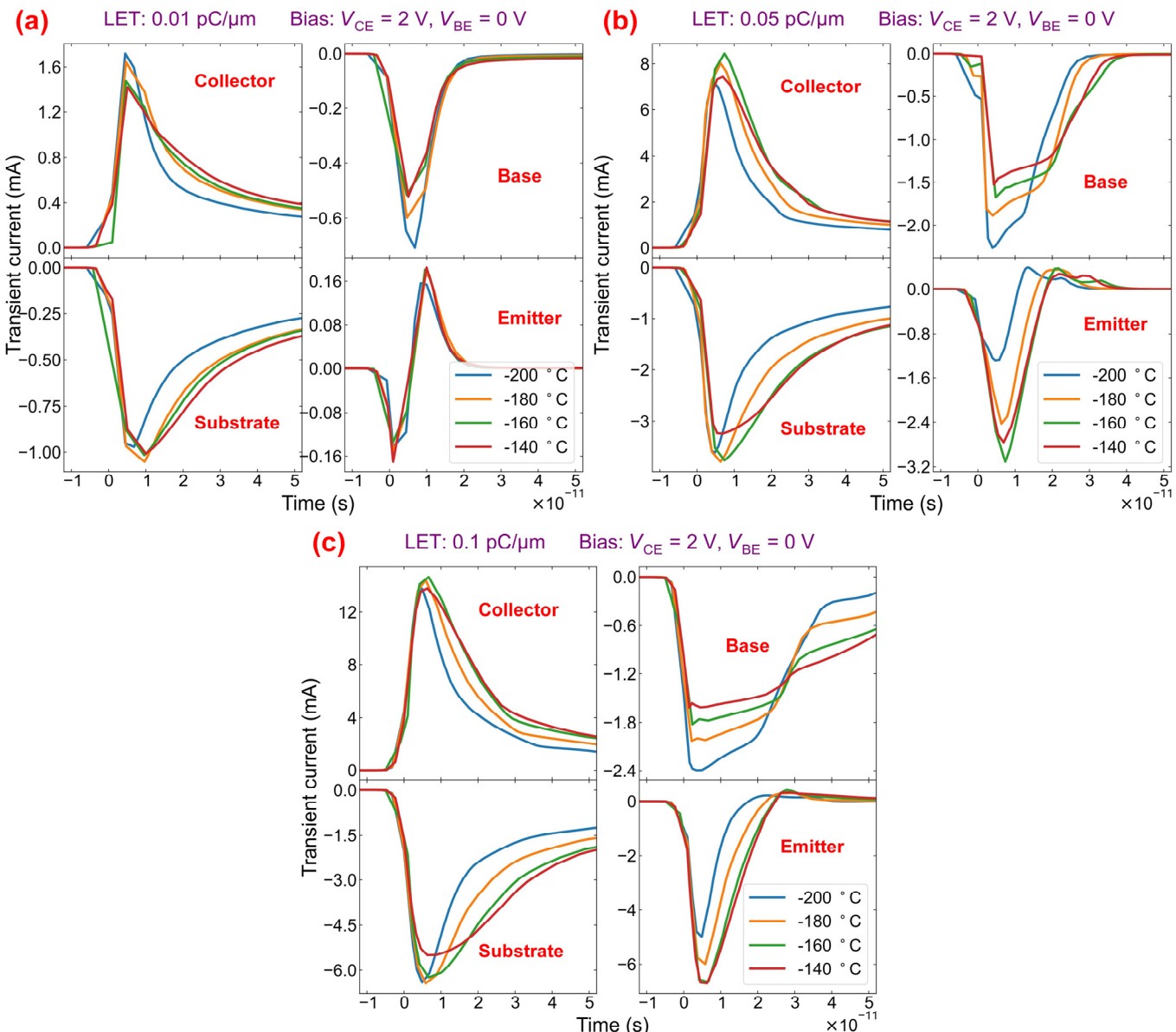

**Figure 4.** The transient current of the four device electrodes from the TCAD simulation at cryogenic temperatures when the LETs are (**a**) 0.01 pC/µm, (**b**) 0.05 pC/µm and (**c**) 0.1 pC/µm, respectively.

Firstly, when the LET is 0.01 pC/µm as in Figure 4a, the ion shunt effect will not be triggered which can be recognized by the weak transient peak of the emitter. At this point, the collector's transient peaks have continued to increase and not shown an inflection point with the temperature decreasing.

As a comparison, when the LET values are 0.05 pC/µm and 0.1 pC/µm, as in Figure 4b,c, the high-density ionized EHPs can connect the emitter and the collector, and there will be a

lot of electrons transferred directly from the emitter to the collector (or the ion shunt effect is triggered on). At this time, the collector's transient peaks will have an obvious inflection point around −160 °C.

That is to say, when the initial ionized EHPs by incident heavy ions are high enough to trigger the ion shunt effect, the inflection point will come earlier.

For a clear analysis, we plotted the transient peaks of the four device electrodes at cryogenic temperatures as in Figure 5. As is generally known, the transient current of the collector could be the sum of the other electrodes, as in (1) [15],

$$i_{cn} = -(i_{bp} + i_{sp} + i_{en}) \tag{1}$$

where $i_{cn}$, $i_{bp}$, $i_{sp}$, and $i_{en}$ represent the transient currents of the collector, base, substrate and emitter, respectively. In addition, the subscript $n$ indicates "electron collection" and $p$ indicates "hole collection". It is not surprising that the sum of all the electrodes' currents should be zero. From Figure 5, we can extract three key features:

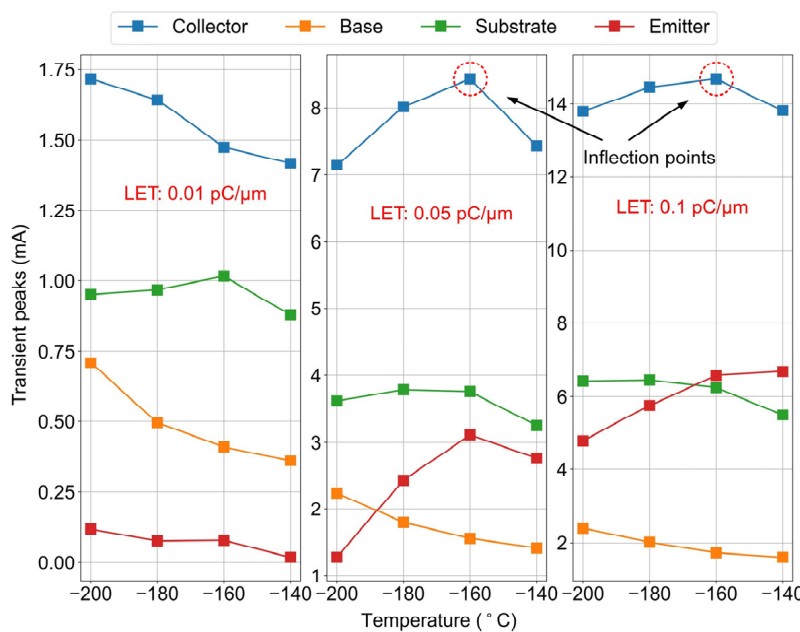

**Figure 5.** The transient peaks (absolute values) of the four device electrodes from the TCAD simulation at cryogenic temperatures with different LET values.

First, whether the ion shunt effect is on or not can be directly reflected by the share of the emitter transient peak (the red lines and squares), which will increase with the LET value. Additionally, when the ion shunt effect is on (LET values are 0.05 pC/µm and 0.1 pC/µm), there will be an inflection point (about −160 °C ~ −140 °C) of the emitter transient peaks. That is to say, the relatively high LET values lead to earlier inflection points. At this time, the total parasitic resistance on the emitter "electron collection" path includes the heavy doping $R_E$, moderate doping $R_{B-V}$, light doping $R_C$ and heavy doping $R_{SC}$ in Figure 3. At cryogenic temperatures, the $R_{B-V}$ and $R_C$ increase as the temperature decreases. In contrast, the $R_E$ and $R_{SC}$ decrease with the temperature. The presence of these two competitive mechanisms together leads to the inflection point. In the future, the total parasitic resistance at a specific temperature will need to be calculated by 3-D simulation.

Second, the base transient peak (the orange lines and squares) continues to increase as the temperature decreases, regardless of the LET value. This is due to the high doping concentration in the intrinsic and epitaxial base regions ($R_{B-L}$) which means that the impurities are almost completely ionized even at a cryogenic temperature. At this time, the parasitic resistance on the base "hole collection" path is mainly influenced by the carrier mobility.

Third, the substrate transient peaks (the green lines and squares) have shown a very different pattern from the emitter. As we can see, the relatively low LET values lead to earlier inflection points. The total parasitic resistance on the substrate "hole collection" path is dominated by the lightly doped substrate and C/S junction regions. When the LET value is 0.01 pC/μm, the heavy ion-induced EHPs' density is relatively low, and the parasitic resistance is controlled by the intrinsic impurity ionization rate and carrier mobility. At this time, the i.i. of the impurities at low temperatures will lead to the peak inflection point (about −160 °C). In contrast, when the LET value is relatively high (such as 0.05 pC/μm and 0.1 pC/μm), the heavy ion-induced EHPs' density will also modulate the parasitic resistance. In extreme cases, the total resistance will be completely taken over by the initial EHPs.

So far, we have found that the inflection point of the collector transient peaks occurs as a combined result of the temperature dependence of the parasitic resistance on the above three carrier collection paths. Furthermore, we can obtain the key conclusion that if we want to conduct a ground-based simulation experiment (typically high LET values), cooling down with the liquid nitrogen (−196 °C) can already meet the requirements.

## 3. Ion Track Simulation by Geant4

### 3.1. Initial Ion Track Structures

In the space radiation environment, heavy ions' energy could even reach hundreds of GeV per nucleon (GeV/amu) and the peak flux is around hundreds of MeV/amu. However, for the ground SEE testing facilities in the world, the heavy ion beam's energy could not exceed 100 MeV/amu [16,17]. As is generally known, the same ion at different energies will have different LET values or different ion track structures; thus, the heavy ion-induced initial EHPs' distribution will be different.

In this section, we will take the typical heavy ion (Fe) in space as an example and study the impact of ion energy on the SiGe HBT's SET inflection point.

The ion track structure was obtained by Geant4 (version 10.7) Monte Carlo simulation [18]. In each simulation round, we simulated the 1000 normally incident Fe ions with energies of 100 MeV, 1 GeV and 10 GeV, respectively. As is shown in Figure 6, the target material is silicon and the ionization energy deposition is counted in the cylindrical coordinate system, because the radial distribution is approximately axisymmetric about the Z axis. Due to the relatively large feature size, the radial spacing and the axial spacing are set to 10 nm and 1 μm, respectively. Furthermore, the calculation accuracy can be further improved by reducing these spacings.

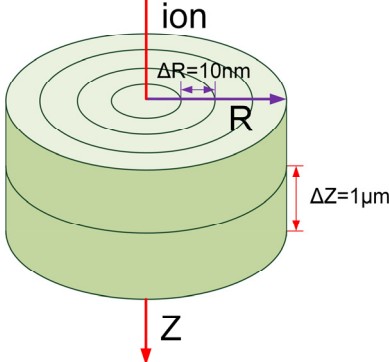

**Figure 6.** The schematic diagram of the cylindrical coordinate system used in the Geant4 simulation.

We could then obtaint the e-h pairs' distribution (shown in Figure 7) by considering the average ionization energy 3.6 eV in silicon. From Figure 7, one could better visualize the differences in the EHPs' distribution generated by ions with different energies. As we can see, the 100 MeV Fe ion's incident depth is about 20 μm and its energy loss is limited in a relatively narrow radial distance. With the increase in ion energy, the ion's incident

depth becomes larger and the EHPs' distribution can reach further radial distances. The EHPs' peak densities induced by Fe ions with energies 100 MeV, 1 GeV and 10 GeV are about $5.25 \times 10^{21}$ cm$^{-3}$, $2.25 \times 10^{21}$ cm$^{-3}$ and $3.59 \times 10^{20}$ cm$^{-3}$, respectively.

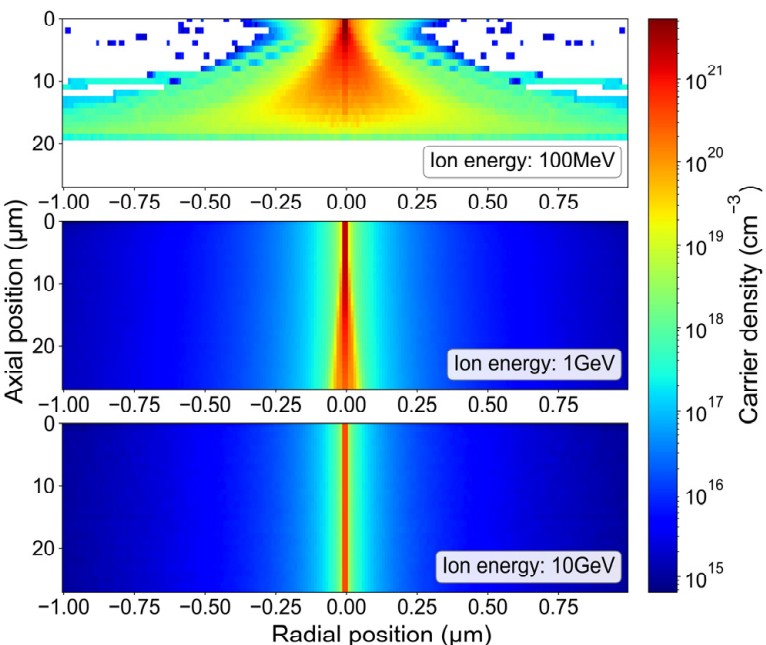

**Figure 7.** The 2-D profile of EHPs' distribution when Fe ion strike in silicon. (Note: the ion energies are 100 MeV, 1 GeV and 10 GeV, respectively.)

According to our previous study [14], the effective charge collection depth of the DUT is 20 µm; therefore, we should pay more attention to this distance. When the ion energy increases from 100 MeV to 10 GeV, the surface LET value decreases from about 0.31 pC/µm to 0.02 pC/µm.

*3.2. Embedding Ion Track to TCAD Simulation*

The most popular method to simulate the heavy ion-induced EHPs' distribution is the Gaussian distribution function in TCAD simulation. In general, the characteristic distance is a constant value and the LET value could be constant or be a function of incident depth. However, this is a simplified empirical model, and some details will be lost.

In the literature [19], the double Gaussian-fitted model is proposed to simulate the heavy ion-induced SEE in the TCAD toolkit, while the accuracy of the simulation is better than the simplistic Gaussian model as mentioned earlier. However, we need to spend a considerable amount of time manually fitting the necessary parameters at different incident depths to make the carrier density distribution as close as possible to the results of the Monte Carlo calculation.

In this paper, we chose a more direct method by defining the spatial distribution function (SDF) in TCAD simulation. As is shown in Figure 8, we should first define the heavy ion and target material in the Geant4 project. We then need to code the C++ script to read the energy deposition results from the Geant4 calculation and build the SDF to extract the corresponding carrier densities according to the different spatial locations, and then call this SDF function in the TCAD command file. Using this method, we could introduce the EHPs' distribution generated by Geant4 to the TCAD simulation directly.

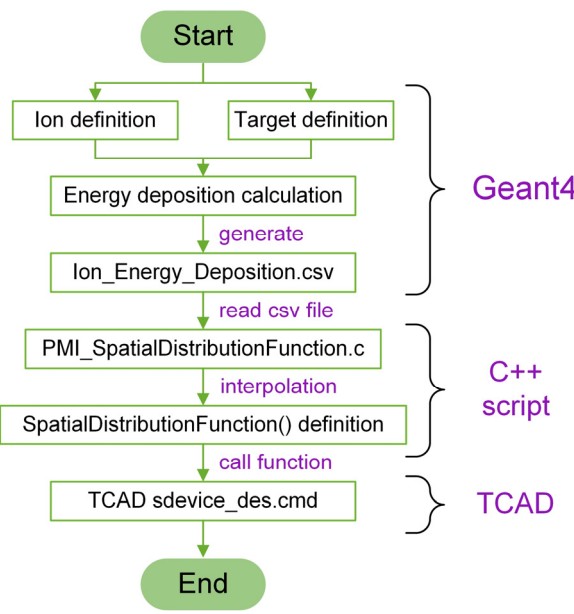

**Figure 8.** The workflow for introducing the EHPs' distribution generated by the Geant4 calculation to the TCAD simulation.

### 3.3. Simulation Results

According to the method in Sections 3.1 and 3.2 we introduced the EHPs' profiles of Fe ions with different energies to the 3-D TCAD process model, as in Figure 9. Crucially, we also need to optimize the meshing strategy to make the EHPs' profiles in the TCAD model and the Geant4 simulation results almost identical. Specifically, we need to use a tighter meshing (about 10 nm) in the central axis of the incident position, especially in the sensitive volumes such as the E/B/C stack structure and the junction regions.

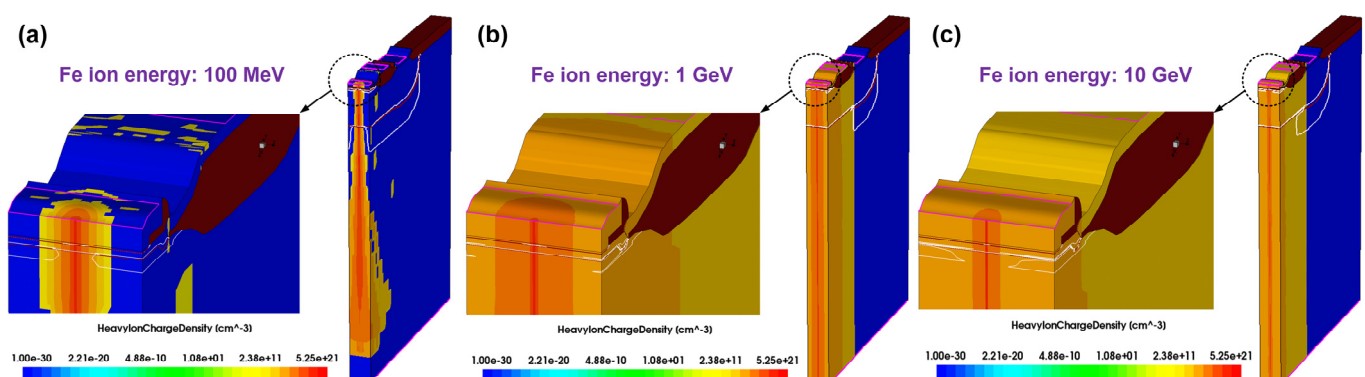

**Figure 9.** The EHPs' distribution of the TCAD simulation produced by incident Fe ions of different energies (**a**) 100 MeV, (**b**) 1 GeV and (**c**) 10 GeV. For a clearer comparison, the charge densities are adjusted to the same scale range.

We could then achieve the SET waveforms of the 3-D process simulation, as in Figure 10. To save simulation time, we have built half of the 3-D model and set the thickness in the z-direction to 1.5 μm. We have also chosen the emitter center normal incidence at room temperature. We could then obtain the information below.

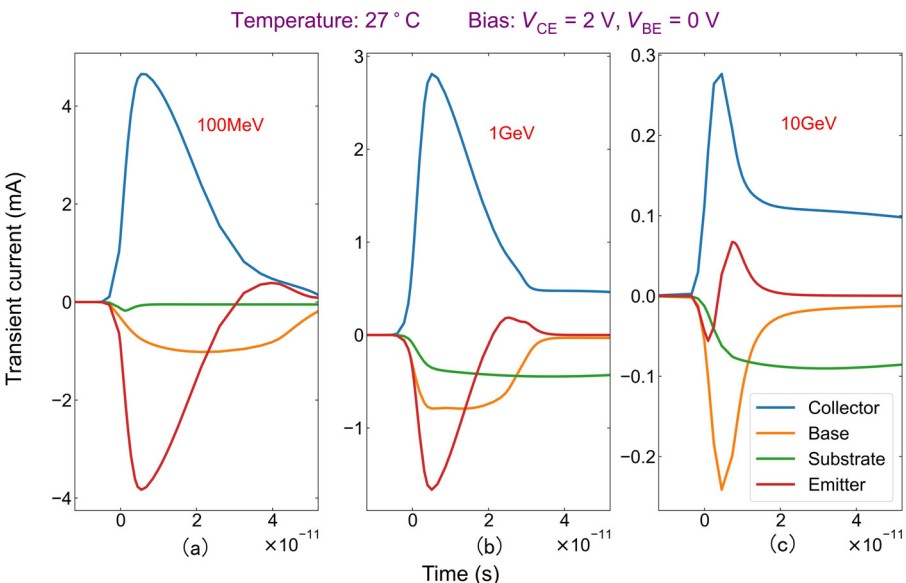

**Figure 10.** The transient waveforms of the TCAD simulation produced by incident Fe ions of different energies (**a**) 100 MeV, (**b**) 1 GeV and (**c**) 10 GeV at room temperature.

First, we are referring to the 100 MeV Fe ion incidence in Figures 9a and 10a. At this time, the Fe-induced EHPs have a peak density about $5.25 \times 10^{21}$ cm$^{-3}$ which is very high to cause the strongest emitter transient current and finally the strongest collector transient current (the transient peak could exceed 4 mA). However, the EHPs' lateral distribution distance is the smallest, and the total EHPs decrease rapidly with the depth of incidence. That is exactly why the substrate transient is much weaker than the emitter transient. Therefore, the collector transient's inflection point is dominated by the emitter transient.

Second, we will focus on the 1 GeV Fe ion incidence in Figures 9b and 10b. At this point, the EHPs' peak density is about $2.25 \times 10^{21}$ cm$^{-3}$ and the emitter transient is weaker than the 100 MeV Fe ion's case. However, the EHPs' lateral distribution distance is much larger, and the total EHPs stay almost constant along the incident depth. The ion shunt effect is still turned on and the share of the substrate transient peak increases significantly. The collector transient is then dominated by both the emitter transient and the substrate transient.

Third, when it comes to the 10 GeV Fe ion incidence in Figures 9c and 10c, the collector transient is the weakest. At this point, the incident Fe ions could deposit energy to the deepest distance. However, the EHPs' peak density is only about $3.59 \times 10^{20}$ cm$^{-3}$ and the ion shunt effect is turned off. According to the results in 2.3, it is hard to see the collector's transient inflection point at a cryogenic temperature.

In summary, the heavy ion's parameters will also have a significant impact on the SET's inflection point at cryogenic temperatures. Added to which, the conventional simulations of heavy ion-induced SEE generally set some fixed model parameters, which can help us to qualitatively analyze the experimental phenomena. However, some EHPs' distribution details are lost and this can affect the accuracy of our analysis. In particular, as the device feature size continues to decrease, the single heavy ion's incidence can affect the operation of more than one device. In this case, our proposed method could help to preserve as much detail as possible about the distribution of the initial ionized EHPs and obtain more accurate simulation results.

## 4. Conclusions

This paper presented an investigation into the inflection point of the single-event transient in a SiGe HBT at a cryogenic temperature. We focused on the impact of the heavy ion-induced initial EHPs' distribution on the inflection point by TCAD simulation. The

collector's transient inflection point is jointly determined by the transient current of the emitter, substrate, and base. Moreover, the characteristics of the transient peaks with the temperature vary greatly among electrodes. The ions with high LET values will trigger the ion shunt effect which can lead to an earlier inflection point (about $-160\,^\circ$C). And when the incident ion's LET value is too low to trigger the ion shunt effect, we will not see an inflection point of the collector's transient inflection point even at $-200\,^\circ$C.

In addition, we proposed a method to directly introduce the initial ionized EHPs' distribution of the Geant4 simulation to the TCAD simulation which could improve the simulation accuracy and efficiency of the heavy ion-induced SEE.

However, the problem of how to improve the convergence of the TCAD 3-D simulation at cryogenic temperatures still remains to be solved and more efforts will be needed in the future.

**Author Contributions:** Conceptualization, X.P. and Y.L.; methodology, X.P. and C.L.; validation, H.Z., C.L. and X.P.; formal analysis, X.P.; investigation, X.P.; resources, X.P.; data curation, H.G.; writing—original draft preparation, X.P.; writing—review and editing, X.P. All authors have read and agreed to the published version of the manuscript.

**Funding:** This research was funded by the National Natural Science Foundation of China (Grant Nos. 61704127, 12005159, and 11775167).

**Acknowledgments:** The authors would like to thank Wen Zhao, Xinshuai Jiang and Yuanyuan Xue for their valuable discussion on Geant4 simulation.

**Conflicts of Interest:** The authors declare no conflict of interest.

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
