# Peer review of "The Inflection Point of Single Event Transient in SiGe HBT at a Cryogenic Temperature"

_electronics, doi:10.3390/electronics12030648_

Round 1

Reviewer 1 Report

Dear colleagues,
In this manuscript, the authors experimentally demonstrated that there will be an inflection point of single event transient in the silicon-germanium heterojunction bipolar transistors (SiGe HBTs) with decreasing temperature from +20 ℃ to -180 ℃. They have found that the incident heavy ion’s parameters could also have an important impact on the this SET inflection point by introducing the ion track structures generated by Geant4 simulation to the TCAD transient simulation.

 The results are interesting. The figures reflect the results of the study. Despite the very good impression of the article, there one remark which could improve the article in my opinion, partly:
As minimum short discussion of obtained results is absolutely necessary.

In summary, I have been satisfied with the high level of the article. I believe this manuscript will attract significant attention from the research community. In my personal opinion, the article is very valuable, a great prospect for further research, and, after minor corrections, can be recommended for publication.

Author Response

We appreciate your valuable suggestions. And we have added some necessary discussion of the obtained results in the revised manuscript.

Reviewer 2 Report

The paper entitled "The inflection point of single event transient in SiGe HBT at cryogenic temperature" simulates the incident heavy ion’s parameters on the SET inflection point.

The scientific content of the work is clear and well detailed. The paper should be published after some minor modifications

Those are some points of improvement:

-     The experimental SET inflection depicted from previous paper could be illustrated here. It would help the readers to get a better understanding of the content of the present paper.

-        More details about the DUT (nature of materials, structure, conditions of fabrication…) are welcome.

-        The inset texts in figure are too small. Please increase the font.

Author Response

Thank you for your very professional and specific comments. We have listened carefully to your comments and made improvements in the revised manuscript. One particular point to note is that more details of the DUT manufacturing process are not shown in the paper because of the commercial confidentiality requirements.

Reviewer 3 Report

I have the following more significant comment to the authors:

The research is interesting and important. My major question (and suggestion) is: is it not possible to perform a special experimental design (DoE procedure) where the output functions transient peaks or transient currents to be modeled with respect to time and temperature simultaneously. It could lead to creation of model  counting for the effects of the independent variables and their simultaneous influence. Further, an adequate model could predict the system behavior for specific conditions.

Author Response

Thank you for pointing out this problem in our manuscript. In our previous study, we used the pulsed laser testing to obtain the SiGe HBT’s transient characteristics at cryogenic temperatures. We could only capture the transient waveform at a specific position and a specific temperature. And the trigger level of the oscilloscope is kept constant during the experiment. We have focused on the most sensitive position (the emitter center incidence) and the only variable that we could change is the temperature. In the TCAD simulation, we strive to simulate the state of the system closest to the real space environment. However, how to improve the convergence of the TCAD 3-D simulation at cryogenic temperatures still needs to be solved and more efforts are needed in the future.